# Workplace Violence in Chinese Hospitals: The Effects of Healthcare Disturbance on the Psychological Well-Being of Chinese Healthcare Workers

**DOI:** 10.3390/ijerph16193687

**Published:** 2019-09-30

**Authors:** Nan Tang, Louise E. Thomson

**Affiliations:** Division of Psychiatry and Applied Psychology, School of Medicine, University of Nottingham, Jubilee Campus, Nottingham NG8 1BB, UK

**Keywords:** workplace violence, healthcare disturbance, emotional labour, psychological well-being

## Abstract

Healthcare disturbance is a form of workplace violence against healthcare workers perpetrated by patients, their relatives, and gangs hired by them. It is a prevalent phenomenon in China, where evidence suggests that it impacts on the job satisfaction of healthcare workers. This study aims to examine the relationship between healthcare disturbance, surface acting as a response to emotional labour, and depressive symptoms in Chinese healthcare workers. The study adopted a cross-sectional design and used an online survey methodology. Data were collected from 418 doctors and nurses from one hospital in China. The results showed that frequency of healthcare disturbance was positively related to surface acting and depressive symptoms, respectively; surface acting was also positively related to depression, while deep acting showed no effect on symptoms of depression. Furthermore, surface acting in response to emotional labour mediated the relationship between healthcare disturbance and depressive symptoms. The results highlight the importance of preventing healthcare disturbance and of training healthcare staff in strategies for managing emotional demands in reducing depressive symptoms in Chinese healthcare staff.

## 1. Introduction

Workplace violence is any “incident where staff are abused, threatened, or assaulted in circumstances relating to their work, including commuting to and from work” [1]. It is categorized into two forms: physical and non-physical violence [2]. Physical violence consists of pushing, kicking, slapping, grabbing, and other forms of intentional physical contact, while non-physical violence includes verbal violence and sexual harassment [3]. 

In China, there is a specific form of workplace violence called ‘Yi Nao’, which translates as healthcare disturbance [4]. This is defined as violence against healthcare facilities and healthcare staff aimed at achieving financial benefits [5]. It is usually perpetrated by patients and their family members or criminal gangs hired by them, with the purpose of forcing the hospital administrations to pay compensation for perceived malpractice [5]. In studies, healthcare disturbance has been classified into three categories: verbal violence (including insults, vulgarity, sarcasm, shouting, etc.); physical violence (including punching, slapping, kicking, and other forms of physical assault); and sexual harassment (including unwanted sexual advances or attention) [6].

Healthcare disturbance is not a recent phenomenon in China. In a study of 270 hospitals in 2006, 70% of healthcare professionals reported that they had experienced Yi Nao events [7]. However, it is becoming more prevalent and the number of healthcare staff who are assaulted, injured, even murdered by patients, visitors, and gangs is increasing [8]. A survey of over 3000 physicians in China found that 76.2% reported exposure to verbal abuse, 58.3% reported that patients ‘made difficulties’ for them, whilst 40.8% said that patients attempted to smear their reputation, 40.2% had experienced mobbing behaviour, 27.6% had experienced intimidation behaviour, 24.1% experienced physical violence, and 7.8% reported sexual harassment [3]. There are also concerns that the deterioration in healthcare staff–patient relations may affect social stability and social harmony as well as leading to lower quality healthcare and inefficiency in the healthcare industry [9,10], which would increase healthcare expenses and worsen patient outcomes. 

Work in the healthcare sector is inherently demanding and requires a high degree of emotional labour—the process of managing feelings in order to achieve the emotional requirements of a job [11]. Healthcare staff are required to regulate their emotions during interactions with patients and, in particular, to show empathy during their interaction [12]. The deterioration in patient–healthcare staff relations in China epitomized by healthcare disturbance [13] places additional pressures on healthcare staff and is likely to increase the emotional labour required of them. This study sought to examine some of the potential consequences of healthcare disturbance on the psychological well-being of healthcare workers and the role of strategies for responding to emotional labour in this relationship.

Healthcare disturbance has specific adverse impacts on both healthcare workers and hospitals, frequently causing injury to staff, physical damage to hospital buildings and property, and also inconvenience and anxiety to other patients. These violent incidents are likely to exacerbate the risk of poor occupational health outcomes of healthcare staff including burnout, work-related stress, emotional exhaustion, depressive symptoms, and anxiety disorders [3,14]. Previous studies have shown a higher prevalence of symptoms of depression in physicians and nurses exposed to physical violence in Chinese hospitals (71.25%) compared to those who have not experienced workplace violence (57.2%) [15]. Moreover, it is reported that those who have specifically experienced healthcare disturbance suffered from depression (28.13%), anxiety (25.67%), and post-traumatic stress disorder symptoms (28%) [16]. Therefore, the following hypothesis is proposed:

**H1:** *Healthcare disturbance is positively related to depressive symptoms*.

Emotional labour is the process of regulating emotions and expressions in the workplace in order to create a positive impression on people to meet job requirements [17,18,19]. There are many public-facing, service sector roles that are identified as requiring emotional labour in the workplace [20]. High levels of emotional demands and stress are reported in these workers, especially healthcare workers [21], and it is likely that the emotional labour of experiencing healthcare disturbance is particularly high. Several studies indicated that emotional demands in the workplace may lead to symptoms of depression such as hopelessness, suicide ideation, chronic sadness, and sleep disturbance [22,23]. In hospitals, nurse and physician roles are emotionally complex occupations involving emotional labour from the efforts required to recognize the emotions of others and to manage their own emotions. The demands of emotional labour occur when an employee has to change his/her actual emotions so that they present emotions that conform to the rules and expectation of the job [24]. In such situations, individuals adopt surface acting, when employees display the emotions required or expected of their role, but their true feelings remain unchanged and are incongruous with their displayed emotions [25,26]. In contrast, deep acting is when an individual will make conscious effort to change their feelings and thoughts towards others [27]. People in jobs with high emotional labour who use these different strategies will experience different consequences [28]. The objective of both surface and deep acting is to show positive emotions, which are presumed to impact the feelings of customers or patients, yet research has shown surface acting to have a negative impact on employee well-being [29]. It has been argued that engaging in surface acting needs higher levels of self-control and consumes more personal resources than engaging in deep acting [30,31] potentially generating negative impacts on emotional labourers’ mental health. Thus, surface acting has been found to be significantly associated with absenteeism, burnout, and depressive symptoms in nurses [32]. Accordingly, the following hypotheses were tested: 

**H2:** *Healthcare disturbance is positively related to surface acting*.

**H3:** *Surface acting is positively related to depressive symptoms*.

In healthcare settings, the effort of emotional labour among physicians and nurses is often experienced when managing the negative outcomes with patients, which will include disagreements, threats, verbal violence, and even physical violence. Emotional labour is the key to make patients feel safe and comfortable [33], and even if they are facing aggressive patients, physicians and nurses are required to regulate their emotions to display a desirable image of their positions and provide good-quality services. In order to meet the needs of improving doctor–patient and nurse–patient relationships, nurses and doctors are likely to experience an increase in the demands of emotional labour. In addition, the increased psychological demands of surface acting in response to the emotional labour faced by them would exacerbate the risk of depressive symptoms. Thus, it follows that healthcare disturbance would increase the demand of emotional labour among healthcare workers and this, in turn, might lead to higher levels of depressive symptoms. Therefore, the following hypothesis is proposed:

**H4:** *Surface acting mediates the relationship between healthcare disturbance and depressive symptoms*.

## 2. Method

### 2.1. Sample and Procedure

Participants were recruited from a Chinese public hospital in July 2018. The hospital is a large paediatric hospital in a first-tier city with over 1200 inpatient beds and over 1800 staff. It provides comprehensive healthcare and medical services for the local population from birth to 18 years old. The sample included both physicians and nurses from different departments within the hospital. Participants were over 18 years of age and had worked in the hospital for at least one year. 

Ethical approval for the study was obtained from the University of Nottingham’s Division of Psychiatry and Applied Psychology Ethics Committee (Ref. 0115). A survey was distributed to 500 nurses and physicians using WeChat online surveys. 433 responses were collected, giving a response rate of the study of 87%. In total, 418 valid responses were completed and included in the analysis.

### 2.2. Measures

A cross-sectional design was adopted using an online survey to collect data on the following sets of variables. Demographic information was collected on age, gender, education level, marital status, job role (i.e., nurse or physician), number of years of employment, number of hours worked per week.

Frequency of experiencing different types of healthcare disturbance was collected based on three categories: verbal violence, physical violence, and sexual harassment. Respondents were asked to report the frequency of these different types of healthcare disturbance they had experienced over the past 12 months. A total score of healthcare disturbance was obtained by summing the scores of the three categories. 

Emotional labour was measured using 11 items from the Emotional Labour Scale (ELS) (Chinese version) [17]. The initial scale consists of 14 items, of which seven items measure surface acting, four items measure deep acting, and three items measure the expression of naturally felt emotions [34]. The ELS adopts a five-point Likert scale, with “never”, “rarely”, “occasionally”, “frequently”, and “always” response options. Higher scores indicate a greater level of surface acting and deep acting. The present study adopted 11 items that measured surface acting and deep acting. The Cronbach’s α value of surface acting and deep acting in this study were 0.85 and 0.70, respectively. Sample items for surface acting are, “I put on an act in order to deal with patients in an appropriate way”, ”I just pretend to have the emotions I need to display for my job”; and for deep acting, “I try to actually experience the emotions that I must show to patients’, “I make an effort to actually feel the emotions that I need to display towards others’.

Depressive symptoms were measured using the Zung Self-Rating Depression Scale (SDS) (Chinese version) [35]. The SDS has good reliability, and it is widely used to measure the level of depressive symptoms related to workplace violence. The scale has 20 items, with a four-point scale ranging from 1 (absent or minimum) to 4 (high or extremely high) [15]. Higher scores indicate higher levels of depressive symptoms. The Cronbach’s α value of depressive symptoms in the study was 0.86.

### 2.3. Analysis

The analysis of the data was carried out in the IBM SPSS Statistics (version 24). The Spearman rank-order correlation coefficient was used to examine the correlations among sociodemographic characteristics, healthcare disturbance, emotional labour (i.e., surface acting and deep acting), and depressive symptoms. Hierarchical multiple regression analysis was employed to examine the relationship between healthcare disturbance, surface acting, and depressive symptoms. Assumptions of the analysis were tested. 

## 3. Results

### 3.1. Sample Characteristics

Table 1 presents the frequency distributions for the demographic information (age, gender, education, marital status, position, years of employment, hours worked per week) of healthcare workers in the Chinese hospital. The sample consisted of 418 respondents, their age ranged between 19 and 60 years old (*M* = 30.40; *SD* = 7.39), with 351 females (84.0%), and 67 males (16.0%). 

The majority of the sample had the job role of nurse (*n* = 306, 73.2%), whilst physicians accounted for 26.8% (*n* = 112). Most of the respondents had worked less than 10 years (*n* = 276, 66.0%), and worked between 40 and 49 hours each week (*n* = 268, 64.1%). In terms of experiencing healthcare disturbance, 37.5% (*n* = 157) had no experience, 51.0% (*n* = 214) had experienced healthcare disturbance 1–4 times in the last 12 months, 7.9% (*n* = 33) had experienced healthcare disturbance 5–8 times, and 3.6% (*n* = 15) reported more than eight relevant experiences. 

Among those who faced some healthcare disturbance in the last 12 months (*n* = 261), 99.6% (*n* = 260) experienced verbal violence, 8.4% (*n* = 22) experienced physical violence, and 1.1% (*n* = 3) experienced sexual harassment. 

### 3.2. Association between Emotional Labour and Depressive symptoms with Sociodemographic Variables

A series of Spearman rank-order correlations were conducted in order to determine whether there were correlational relationships among sociodemographic variables, healthcare disturbance, surface acting, deep acting, and depressive symptoms. Table 2 shows the means, standard deviations, correlations, and reliability coefficients of the variables. In terms of reliability coefficients, all variables show an acceptable Cronbach’s α [36].

Regarding sociodemographic variables, results indicated that job role showed statistically significant but small associations with surface acting (r = 0.149, *n* = 418, *p* = 0.002), deep acting (r = 0.109, *n* = 418, *p* = 0.026), and depressive symptoms (r = −0.225, *n* = 418, *p* < 0.001), so nurses showed higher levels of surface acting, deep acting, and depressive symptoms compared to physicians; moreover, hours worked per week revealed weak correlations with surface acting (r = 0.128, *n* = 418, *p* = 0.009) and depressive symptoms (r = 0.163, *n* = 418, *p* < 0.001). 

According to the results of the present study, there were positive correlations between healthcare disturbance, surface acting, deep acting, and depressive symptoms. Firstly, healthcare disturbance was positively correlated with surface acting (r = 0.284, *n* = 418, *p* < 0.001), deep acting (r = 0.163, *n* = 418, *p* < 0.001), and depressive symptoms (r = 0.214, *n* = 418, *p* < 0.001). These results confirmed H1 and H2. Secondly, surface acting was positively correlated with deep acting (r = 0.348, *n* = 418, *p* < 0.001) and depressive symptoms (r = 0.166, *n* = 418, *p* < 0.001). However, there was no statistically significant correlation between deep acting and depressive symptoms (r = −0.037, *n* = 418, *p* = 0.45). The results indicate that regarding the relationships between emotional labour and depressive symptoms, surface acting showed a significantly positive association with depressive symptoms, while there was no association between deep acting and depressive symptoms, which supported H3.

### 3.3. Predicting Depressive symptoms from Healthcare Disturbance and Surface Acting 

From the results in Table 2, work-related factors (job role and work hours per week) showed statistically significant relationships with both surface acting and depressive symptoms. Hence, job role and number of hours worked per week were incorporated as predictive variables when employing hierarchical multiple regression.

Prior to conducting a hierarchical multiple regression, its relevant assumptions were tested. Firstly, the analysis program G*Power 3.1 calculated the estimated minimal sample size as 85, by using linear multiple regression (fixed model, R^2^ deviation from zero). The effect size was set as 0.15, and the statistical power was set as 0.80. In the present study, a sample size of 418 was deemed adequate given four predictors to be included in the analysis. Secondly, the assumptions of linearity and unusual cases were satisfied. Furthermore, the assumptions of multivariate normality, multicollinearity, and homoscedasticity were deemed to have been met.

A two-stage hierarchical multiple regression was conducted to determine whether or not healthcare disturbance and surface acting contributed incrementally to the prediction of depressive symptoms above and beyond that accounted by the job role and the work hours per week. Job role and work hours per week were entered in step 1, then healthcare disturbance and surface acting were entered in step 2. Results indicated that work-related variables explained 8.4% of the variance, (F(2, 415) = 19.14, *p* < 0.001). Furthermore, healthcare disturbance, surface acting, and work-related variables together explained 14% of the variance in depressive symptoms, (F(2, 413) = 14.43, *p* < 0.001). Partial regression coefficients are reported in Table 3. 

As illustrated in Table 3, all predictors accounted for a significant proportion of unique criterion variance in the final regression model. These indicate that in model 2, job role, hours worked per week, healthcare disturbance, and surface acting were good predictors of depressive symptoms. Specifically, regarding to job role, *B* was −4.29, which indicated that compared with physicians, nurses showed higher levels of depressive symptoms. For other independent variables, the higher their levels were, the higher the levels of depressive symptoms would be. 

### 3.4. Mediation Relationship

Based on a simple mediation analysis, healthcare disturbance influenced depressive symptoms through a direct effect on surface acting. As shown in Figure 1 and Table 4, healthcare disturbance also positively predicted surface acting (b = 0.54, *p* < 0.001), and respondents’ surface acting was also positively predictive of depressive symptoms (b = 0.22, *p* = 0.006), results supported the mediational hypothesis. A bias-corrected and accelerated (BCa) confidence interval for the indirect effect (b = 0.12, *p* = 0.014) of healthcare disturbance’s prediction of depressive symptoms through surface acting (based on 1000 bootstrap samples) was entirely above zero (95% CI = 0.03–0.21); this represented a small effect size (k^2^ = 0.0351, 95% CI = 0.01–0.06). 

There was also significant evidence that healthcare disturbance positively predicted depressive symptoms independent of the mediating effect of surface acting (b = 0.67, *p* = 0.001), which supported H4, surface acting mediates the relationship between healthcare disturbance and depressive symptoms.

## 4. Discussion

This study examined the relationships between healthcare disturbance, emotional labour, and depressive symptoms among Chinese nurses and physicians. It was found that 62.5% of the respondents claimed that they have experienced at least one type of healthcare disturbance over the past 12 months, and verbal violence was the most prevalent form of healthcare disturbance, reported by 62.2%. These figures concur with other recent studies, such as [37], whose survey of 15,970 nurses in China reported that 65.8% had experienced work-related violence, with 64.9% having experienced verbal violence, 11.8% physical violence, and 3.9% sexual harassment. As with previous studies, the prevalence of exposure to sexual harassment was the lowest. However, possible under-reporting of sexual harassment may mean that the number of actual cases is larger. Due to cultural factors in China, victims’ experience of sexual harassment may not be disclosed as individuals can consider it a shameful and humiliating experience which would negatively influence how they are viewed by others. A number of reasons for typical non-disclosure of sexual violence amongst Chinese women have been described including embarrassment and social shame, lack of legal and social support, potential social isolation, and family disgrace [38].

In our study, the number of hours worked per week showed significant positive relationships with both aspects of emotional labour (surface acting and deep acting) and depressive symptoms. Longer work hours will increase healthcare workers’ interaction with patients and result in higher levels of emotional demands which require more effort in both surface acting and deep acting. The majority of participants (83.3%) reported that they usually work over 40 hours per week, which is longer than the legal work hours in China. The correlation between longer working hours and depressive symptoms corresponds to the wider literature on the impact of long working hours on poor mental health in healthcare [39,40] and general working populations [41]. Furthermore, without appropriate financial rewards, longer working hours can lead to lower job performance [42]. This could exert adverse impacts on healthcare quality, aggravating the already intense doctor–patient and nurse–patient relationships and further increasing the risks of healthcare disturbance. With regard to job roles, nurses in the hospital showed higher levels of emotional labour and depressive symptoms compared to physicians. Nurse roles typically involve a higher frequency of patient interaction than that of physicians [43]. 

In terms of emotional labour strategies, it is often assumed that surface acting and deep acting are negatively correlated, and that surface acting and deep acting are mutually exclusive, with one strategy used at the expense of the other [44]. On the contrary, the results of the present study indicate that surface acting and deep acting were positively correlated. People who engage in more surface acting also show higher levels of deep acting [45]. This suggest that when managing emotions in the workplace, Chinese physicians and nurses might employ both strategies. Other studies have also supported the positive correlation [29].

In terms of relationships among healthcare disturbance, emotional labour, and depressive symptoms, firstly, the results reveal that frequency of experiencing healthcare disturbance was positively associated with healthcare workers’ depressive symptoms. It is suggested that violence in hospitals could directly aggravate the prevalence of mental health symptoms [46]. Therefore, preventing healthcare disturbance is essential to protect healthcare workers and improve their psychological health.

Secondly, the findings show that higher levels of surface acting could lead to higher levels of depressive symptoms, but deep acting in relation to emotional labour showed no adverse impacts on depressive symptoms. A study conducted among call centre workers [47] also reported different effects of surface acting and deep acting, more specifically, higher levels of surface acting predicted higher levels of depressive symptoms, suggesting that surface acting exerted harmful impacts on psychological health. Moreover, it is suggested that deep acting could improve customer satisfaction [26]. Hence, changing emotional labourers’ coping strategies might be a useful approach to reduce symptoms of depression and increase patient satisfaction, which may, in turn, reduce the number of healthcare disturbance incidents. The above-mentioned information suggests that when managing healthcare workers’ psychological well-being, their experience of healthcare disturbance and their effort of emotional labour needs to be taken into account. 

The study also tested the mediating effects of surface acting on the relationship between healthcare disturbance and depressive symptoms among Chinese nurses and physicians. Results indicate a partial mediating effect, supporting the hypotheses that healthcare disturbance and surface acting might trigger depressive symptoms. This finding indicates that surface acting might compound the psychological burden of healthcare disturbance, thus, aggravating nurses’ and physicians’ depressive symptoms. Therefore, to reduce their depressive symptoms caused by exposure to healthcare disturbance, their effort of surface acting should be reduced. 

### 4.1. Study Implications

Several practical and managerial implications emerge from the study. Firstly, hospitals should adopt a zero-tolerance attitude towards healthcare disturbance [48] and develop a safe work environment for healthcare workers, which provides protection from all forms of work-related violence. In order to maintain positive interactions between staff and patients and also to provide staff with individual strategies for managing difficult interactions with patients and their families, hospitals should put in place training interventions that are focused on communication skills. Evidence suggests that performance-based training with practicing clinicians is particularly effective in improving communication performance [49]. Participatory and experiential approaches to training are most effective, in which case studies and real-life scenarios are used to demonstrate how to respond to patients in different situations with clinicians given opportunities to practice those responses and behaviours [50,51]. Communication skills training should also encourage staff to respond to emotional labour through deep acting, expressing their true feelings instead of faking the desired emotions required by jobs. This might further reduce their depressive symptoms. To do this, enhanced understanding between patients and healthcare workers is required with staff becoming more thoughtful about the needs of patients and their families and adopting a more patient-centred approach to care [52]. Training in the organizational strategies and procedures towards violent incidents should further increase staff confidence in managing violence [53]. However, the organizational approach to preventing and managing healthcare disturbance and other forms of work-related violence needs to go further than this and consider the influence of other aspects of the way work is designed and managed. In our study, working long hours was related to higher levels of emotional labour and depressive symptoms, and this may be a risk factor for some of the conditions that increase the likelihood of healthcare disturbance, such as medical errors of adverse events. It is estimated that 1.6–7.6 million patients are affected by hospital adverse events each year in China [54] and recent studies have highlighted the lack of patient safety culture within Chinese hospitals as well as a punitive approach to responding to medical errors [55]. Long hours in the healthcare sector is also related to poor patient safety culture [54,55,56]. Whilst physicians and nurses clearly work in difficult and challenging environments in China [3,4,55,56], clear leadership and management is required to design systems that benefit both patients and healthcare staff. 

Compared with physicians, nurses showed more serious depressive symptoms. Hence, it may be more necessary to pay additional attention to nurses’ psychological health and use appropriate systems to monitor and manage their workplace health. Hospitals should provide all staff with interventions and organizational support mechanisms that have a positive influence on workers’ psychological well-being and work productivity [57,58]. 

Together, the interventions suggested above support calls for the need for an integrated approach to effectively address workplace mental health, which combines prevention, protection, and treatment approaches [59].

Drawing on the results, the present study offers several contributions to the existing knowledge. Firstly, the study extends recent experimental findings in the Chinese healthcare context. Secondly, different strategies for emotional labour have different effects on psychological well-being, specifically such as depressive symptoms. Thirdly, the study examined the impacts of healthcare disturbance and emotional labour on depressive symptoms simultaneously, broadening the understanding of the cause of depressive symptoms, and providing theoretical evidence for managing depressive symptoms. 

### 4.2. Potential Limitations and Future Directions

Due to the cross-sectional design, we are unable to infer a causal association between healthcare disturbance, emotional labour, and depressive symptoms. Hence, prospective cohort studies could be adopted in future studies to examine whether the changed emotional labour responses would have a significant impact on depressive symptoms. 

Second, the sample of the study was recruited from a particular hospital in one city in China, so its social background and current situation of healthcare disturbance are different from other cities. It is also expected that individuals who had experience of healthcare disturbance were more likely to respond to the survey, leading to self-selection bias. The sample was also heavily weighted by gender, with 84% of respondents being female. Therefore, the results cannot be generalized to healthcare workers (physicians and nurses) elsewhere. Furthermore, compared to the population of Chinese nurses and physicians, the study sample was small, so it is not likely to be representative of the population of Chinese nurses and physicians. Larger samples across multiple sites are needed in future research. 

Third, there are several triggers of emotional labour and depressive symptoms in the healthcare industry, but the present study solely focused on the impact of healthcare disturbance. Hence, there is a need for further research to include other influencing variables as well as additional outcomes such as anxiety, burnout, absenteeism, work performance, and turnover. 

Finally, the study adopted self-reported online questionnaires, so all answers were based on the respondents’ own perception. Hence, there is a possibility that their responses were influenced by social desirability leading to response bias. Related to this, there is a possibility that the data for frequency of experiencing healthcare disturbance might suffer recall bias, because the respondents might have difficulty recalling the exact number of times they had experienced these kinds of violent incidents. 

## 5. Conclusions

This study investigated the relationships between healthcare disturbance, emotional labour, and depressive symptoms among Chinese healthcare workers using an online questionnaire. Results showed that most of the healthcare workers in the study had been victims of healthcare disturbance, with verbal violence being the most prevalent form. Additionally, healthcare disturbance could exert extra burden on the management of emotions in the workplace management, and different emotional management strategies relate differently to depressive symptoms. Specifically, surface acting had adverse impacts on depressive symptoms, while deep acting has no effects on the symptoms of depression. Furthermore, those working longer hours and those in nursing roles showed more severe depressive symptoms. 

It is essential to investigate influential factors of depressive symptoms, such as healthcare disturbance and emotional labour. There are limited studies focused on the impacts of health disturbance on Chinese healthcare workers, mixed method studies could be employed to gain a deeper understanding of the consequences of healthcare disturbance. This can help to inform interventions to prevent healthcare disturbance, to manage healthcare workers’ psychological well-being, and to improve healthcare quality in China. 

## Figures and Tables

**Figure 1 ijerph-16-03687-f001:**
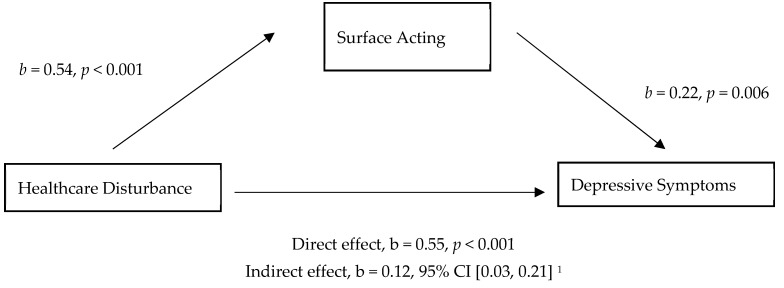
Model of healthcare disturbance as a predictor of depressive symptoms, mediated by surface acting. ^1^ The confidence interval for the indirect effect is a BCa bootstrapped CI based on 1000 samples.

**Table 1 ijerph-16-03687-t001:** Sociodemographic characteristics of the participants (*N* = 418).

Characteristics	Groups	Frequency (*N*)	Percentage (%)
Age	≤29	234	56.0
	30–39	134	32.1
	40–49	39	9.3
	≥50	11	2.6
Gender	Male	67	16.0
	Female	351	84.0
Education	College	117	28.0
	Bachelor’s degree	239	57.2
	Master’s degree	56	13.4
	Doctor	6	1.1
Marital Status	Single	183	43.8
	Married	228	54.5
	Divorced	7	1.7
Job Role	Nurse	306	73.2
	Physician	112	26.8
Years of Employment	≤9	276	66.0
	10–19	107	25.6
	20–29	26	6.2
	≥30	9	2.2
Work Hours per Week	≤39	70	16.7
	40–49	268	64.1
	50–59	54	12.9
	≥60	26	6.3
Frequency of Healthcare Disturbance in the past 12 months	0	157	37.5
	1–4 times	214	51.0
	5–8 times	33	7.9
	≥9 times	15	3.6

**Table 2 ijerph-16-03687-t002:** Respondents reports of demographic variables, healthcare disturbance, surface acing, deep acting, and depressive symptoms: descriptive statistics, correlations, and reliability coefficients.

		M	SD	α	1	2	3	4	5	6	7	8	9	10	11
1	Age	30.40	7.39		-										
2	Gender	1.84	0.37		−0.182 ***	-									
3	Education Level	1.88	0.68		0.570 ***	−0.214 ***	-								
4	Marital Status	1.60	0.59		0.706 ***	−0.139 **	0.395 ***	-							
5	Job Role	1.27	0.44		0.348 ***	−0.339 ***	0.573 ***	0.156 **	-						
6	Years of Employment	8.46	7.40		0.918 ***	−0.103 *	0.409 ***	0.682 ***	0.144 **	-					
7	Hours of Work per Week	43.23	7.64		−0.073	−0.043	0.096 *	−0.058	0.020	−0.133 **	-				
8	Healthcare Disturbance	1.95	2.39		0.020	−0.023	0.086	0.059	0.079	−0.010	0.211 ***	-			
9	Surface Acting	18.72	5.00	0.85	0.069	0.083	0.145 **	0.058	0.149 **	0.027	0.128 **	0.284 ***	-		
10	Deep Acting	12.33	2.88	0.70	0.071	−0.013	0.039	−0.080	0.109 *	−0.047	0.093	0.163 ***	0.348 ***	-	
11	Depressive symptoms	42.37	7.83	0.86	−0.130 **	0.043	−0.040	−0.087	−0.225 ***	−0.099 *	0.163 ***	0.214 ***	0.166 ***	−0.037	-

*N* = 418. * *p* < 0.05, ** *p* < 0.01, ****p* < 0.001.

**Table 3 ijerph-16-03687-t003:** Summary of hierarchical regression analysis for variables predicting depressive symptoms.

Variable	B [95% CI]	Std. Error B	Sr^2^	ß	R	R^2^	ΔR^2^
**Step 1**					0.29	0.08	0.08
Job Role	−4.29 [−5.93, −2.65] ***	0.84	0.006	−0.24			
Work Hours	0.20 [0.10, 0.29] ***	0.048	0.004	0.19			
**Step 2**							
Job Role	−4.85 [−6.45, −3.24] ***	0.82	0.007	−0.28	0.38	0.14	0.136
Work Hours	0.14 [0.05, 0.24] **	0.05	0.002	0.14			
Healthcare Disturbance	0.54 [0.23, 0.85] **	0.16	0.002	0.17			
Surface Acting	0.24 [0.09, 0.39] **	0.08	0.002	0.15			

*Note. N* = 418. * *p* < 0.05, ** *p* < 0.01, ****p* < 0.001.

**Table 4 ijerph-16-03687-t004:** Mediation for surface acting and depressive symptoms with healthcare disturbance as independent variable (IV).

DV	M	Effect of IV on M	Effect of M on DV	Direct Effect	Indirect Effect	Total Effect
DS	SA	0.54 ***	0.22 **	0.55 ***	0.12 *	0.67 ***

*Note.* DV = dependent variable; M = mediating variable; IV = independent variable (healthcare disturbance); DS = depressive symptoms; SA = surface acting. * *p* < 0.05, ** *p* < 0.01, *** *p* < 0.001.

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
