# Peer review of "Workplace Violence in Chinese Hospitals: The Effects of Healthcare Disturbance on the Psychological Well-Being of Chinese Healthcare Workers"

_ijerph, 2019, doi:10.3390/ijerph16193687_

Round 1

Reviewer 1 Report

Include the survey response rate for the surveys, add greater detail on the recruitment of participants and the work setting. 

Include definitions related to the ELS items and sample item(s). 

Expand study implications and support recommendations.

Author Response

We thank Reviewer 1 for their helpful comments for improving our article and have addressed them in the following ways:

1. Include the survey response rate for the surveys, add greater detail on the recruitment of participants and the work setting

These details have been added in line 108-116 of the Methods

2. Include definitions related to the ELS items and sample item(s). 

These details have been added in line 130-136 of the Methods

3. Expand study implications and support recommendations.

The study implications have been expanded quite extensively (line 302-340 in the Discussion) to include further details of types of training, as well as more preventative measures concerning work design and management, including consideration of the patient safety culture.

Reviewer 2 Report

Thank you for the opportunity to review your manuscript on this very important and timely topic. I would recommend removing the "Widow" category from table 1 as there were no responses and this row does not add any value to the information. Further within your limitations section it is important to add self selection bias because this was a survey participants who have experienced healthcare disturbance may have been more likely to respond. 

Author Response

We would like to thank the reviewer for their helpful comments to improve our paper. We have addressed them as follows:

1. I would recommend removing the "Widow" category from table 1 as there were no responses and this row does not add any value to the information.

This line has been removed from Table 1, line 156.

2. within your limitations section it is important to add self selection bias because this was a survey participants who have experienced healthcare disturbance may have been more likely to respond. 

Thank you for pointing this omission out. It has now been added at line 355-256.